# Dynamic reflections of multidimensional health poverty in Pakistan

**Kiran Mustafa[1], Misbah Nosheen[1], Atta Ullah Khan[2]***

**1** Department of Economics, Hazara University, Mansehra, Pakistan, **2** Department of Economics, AIOU, Islamabad, Pakistan

* attaullah.khan@aiou.edu.pk

**Data Availability Statement:** The survey data has been taken by the Federal Bureau of Statistics (FBS), Islamabad; which is an organization of the Government of Pakistan and conduct nationwide survey from time to time. FBS issues data sets to

## Abstract

The recent methodological development has entirely shifted the identification of poor in the multidimensional spectrum; thereby addressing the multiple health spheres. The present research primarily examines the dynamics of multidimensional health poverty on the basis of HIES & PSLM nationwide survey data from 2013–14 to 2018–19. The study employed Alkire & Foster Alkire, S (2007) Multidimensional Poverty Index to estimate the seven distinct dimensions of health aspects to identify the poor. The results of health poverty demonstrate a declining trend over time at national, provincial and regional level in Pakistan. Interestingly, the regional statistics indicated the poverty as a rural phenomenon of Pakistan. Comparative measures of provinces reveal that Baluchistan has been a severe victim of health poverty at overall as well as regional level during the study period. The population decomposition elaborates that individuals residing in two most populated provinces Punjab & Sindh, were the major contributor to overall profile of health poverty. Findings of dimensional decomposition exposes that five key dimensions i.e. use of health services, quality of health services, maternal health, child health and malnutrition have contributed to the overall profile of multidimensional health poverty.

## Introduction

The prevalence of poverty has been a well-known fact over the history of developing nations and universally acknowledged as one of the major restricting factor in materializing the dream of sustainable development. Thus, poverty has played an important role in hollowing the social and economic roots in the emerging economies. Meanwhile, the poverty dilemma has also been discussed in the various forms across the different regimes of time. For example, the conventional discussion on poverty was broadly based on single dimensional concept as insufficiency of income (lesser than $ 1.25/ day) that is essentially required to satisfy rudimentary wants for the survival of life [1, 2].

In the literature, inclusion of Amartya Sen,s capability approach as a key derivative for the welfare masses has entirely shifted the poverty debate from single dimension notion to the multiple attributes; thereby involving the various social and economic aspects [3]. Later, this capability approach has been prolonged as key socio-economic attributes i.e. education, health

the academicians upon the request of research scholars through proper channel for the research purpose. However, recently the data requests can also be made at the web address https://www.pbs.gov.pk/content/data-request-form.

**Funding:** There is no funding source in this study.

**Competing interests:** There is no competing interest.

and housing facilities [4–7]. During the last couple of years, the multidimensional poverty debate has been further proceeded by various scholars, social thinkers and intellectual community in the distinct directions like energy poverty, Child poverty & health poverty [8–10]. Though, all these ideas approximate the magnitude of poverty; but the last one has gained serious consideration because of the fact that over time it shed an overwhelming impact on Asian & African emerging economies, [11].

Although income has been considered as a basic yardstick for the enhancement of living standard of common man [12–14], yet the importance of improved health conditions can never be ignored, particularly in the developing nations [15, 16]. It is mainly due to the fact that increased income proved to be useless when a society is suffering from severe health poverty [17, 18]. Consequently, at one end it is well known fact that the health status discloses a considerable effect on living standard of a common man directly, by persuading its physical and mental functioning [19, 20]. On the other hand, it effects indirectly by limiting the education and financial possessions. For instance, deprived health conditions not only raise the incapability towards getting education but also hamper in attainting the monetary resources through reducing employment level As a result, it becomes imperative to diagnose the responsible factors that exclude the people socially; rather using the end remedies for attaining the sustainable living pattern [21, 22].

Despite the fact that a great number of countries from Asian and African regions had achieved significant reduction in monetary poverty during the last decade, but still there prevails a considerable health deprivation. Meanwhile, most of the countries from these regions have not made any reasonable development in the health sector to lessen basic health problems, especially related to female and children like as pre & post-natal care during pregnancy and infant mortality rates. For instance, WHO [23] elaborated that infant and child mortality at Kenya has been extremely high and estimated numeric reveals that 85 out of 1000 infants born alive and die before reaching of age of 5 years while 16% of children for the age less than 5 are underweighted. Likewise, in South Asian countries average life expectancy has been found as 40 years and 879 million people have poor access to safe sanitation system among these 661 (98) million belong to India (Pakistan). About 278 million people have no access to purified drinking water and 22% of the whole region population has no basic health facility out of which 45% live in Pakistan. While Infant mortality rate in most of the countries of Asian region like Pakistan, Nepal, India, Siri Lanka and Maldives found to be 95, 75, 71, 17 and 53 out of 1000 [11, 24]. Moreover, Awiti [25] illustrated that at the start of the previous decade, number of economies from Sub-Saharan African countries like Kenya, Tanzania, African region, Western Pacific, Europe and America have been facing the substantial infant mortality rate i.e. 8%, 10%, 13%, 5%, 7% and 8% respectively.

Because of the scare literary support regarding the health poverty in the multidimensional context, different scholars & policy makers have indirectly defined the health poverty. For example, Clarke & Erreygers [26] consider it as a state of being in "poor health". Likewise, Arif et al., [27] discussed health poverty through morbidity and malnutrition concept. Similarly, Alderman [28] highlighted that income poverty leads to scarcity of food that causes malnutrition. Meanwhile, World Bank [29] further supported the idea and explored that income poverty and child malnutrition are directly linked to each other. Likewise, Ervin [30] explored that poor child health & nutritional level are the key contributor towards the deprivation of health sector. In addition, Black et al., [31] explained that child health interventions significantly led to decrease the child mortality that ultimately reduces the health poverty. Interestingly, Mohantay et al., [32] identified that unaffordable health care is the key contributor to health sector poverty. Furthermore, several researchers focused on the role of social media in creating health awareness as health campaigns and awareness programs can be disseminated in a very

short span of time. Latha *et al.*, [33] diffusion of information is the key element of creating awareness as highlighted by Lapointe *et al.*, [34]. Alnasser *et al.*, [35] highlighted the positive impact of social media in promoting awareness during Covid-19. In the light of above facts, it has been clearly perceived that the reflections of health sector poverty have crucial importance, especially for the developing economies, but unfortunately no authentic study has been found that explore the particular emerging challenge in a wider & deeper sense especially for developing country like Pakistan. The study in hand is a pioneering effort to explore the dynamics of poverty through a rich variety of health aspects at the national, sub-nationals and regional (rural& urban) level in Pakistan for the last five years. Thus, the present study has made remarkable effort to bridge the gap by estimating health poverty in context of multiple attributes since 2013–14 to 2018–19 in Pakistan. This conceptual shifting give rise to the variety of the questions like; how multidimensional framework is workable for measuring the health attributes? what are the key attributes used to explore health poverty at national, sub-national and regional level in Pakistan? How weighting strategy would be employed for the multiple attributes? What is the trend of multidimensional health poverty at national, sub-national and regional level in Pakistan? How the population proportionate & dimensional contribution to the overall index is meaningful? And so on. In the light of core questions, the current study explores the magnitude of multidimensional health poverty at national, sub-national and regional level in Pakistan. The approximation of heath poverty in the multidimensional spectrum portrays the literary justification while materializing the core objectives of the study. The ccurrent study investigates the decomposition of health poverty in multidimensional viewpoint. In addition, a set of appropriate policy measures have been drawn to eradicate the health poverty in Pakistan.

To accomplish the core objectives of the study, the present research employed the universally acknowledged indexing approach as presented by Alkire and Foster, [4]. The study has been classified into four dissimilar segments. The first part illustrates the brief view of the problem; whereas, the second part critically examines the literature review. Third section, briefly discuss the materials & methods and fourth segment explains the results and discussions. The last section contains the summary and policy implications.

## Literature review

A wide range of literature highlights the fact that assessment of poverty as multidimensional dilemma presents clearer picture of an economy as compared to single dimensional measure. Interestingly, in the recent era, the multidimensional poverty dilemma has progressed to the further distinct categories like health poverty, energy poverty & child poverty. A significant number of studies have been found which describe directly or indirectly the dilemma of health poverty by utilizing divergent ways and tactics.

Wang [36] highlighted that the significant differences in the health facilities are the key factors that causes higher health poverty in rural area as compared to urban counterpart. Thus upsurge in easy availability of necessary health facilities like vaccination, experienced staff and essentially needed medicines in rural areas could be quite helpful for reduction of health poverty. Interestingly, Alkire & Fang [18] argued that poverty is purely a multidimensional phenomenon with highest dimensional deprivation in health sector like lack of access to purified drinking water and safe sanitation system along with high mortality rate & low nutritional level in children and females.

Andersen and Newman [37] explored that distance, time and travel costs toward nearest basic health unit are the major ingredient of health poverty in rural areas of maximum number of developing economies. Likewise, Sasaki *et al.*, [38] demarcated that difference in ecological

locations also played a vibrant role to understand the spread of divergent diseases and dynamic of health poverty in maximum number of emerging economies of the world. Meanwhile, Arif *et al*., [27] highlighted that malnutrition is the key indicator that causes severe health poverty in children of Pakistan.

Edward and Meade [39] globally examined health poverty with three key dimensions i.e. hunger, death and diseases and concluded that hunger and six different types of diseases pneumonia, diarrheal diseases, HIV/AIDS, tuberculosis, malaria, and hepatitis are the key factors that causes health poverty worldwide. Similarly, Shams [40] highlighted health poverty in the light of four very important socio-economic (gender, education, income and age) dimensions. On the other hand, Rehman and Zimmer [41] described child health status by utilizing maternal literacy, poverty, vaccination, level of nutrition & water and Sanitation as key dimensions. Ahmed and Mustafa [42] found that people living in slums face severe health problems.

Ibqal and Nawaz [10] analyzed the profile of multidimensional health poverty at country, provincial and regional level in Pakistan by using well-accepted multidimensional poverty methodology and five key dimensions i.e. use of health services, quality of health services, cost of health services, maternal health and child health. They revealed that overall Pakistan multidimensional health poverty is purely a rural phenomenon. However, provincial measures demonstrated that province Punjab is most while Baluchistan is least affected in context of health poverty. Bakhtiari and Miesami [43] portrayed that improvement in health structure and different educational levels of people are the key factors that led to decline poverty. Similarly, Bloom and Cunning [44] also explored that improved health & human wellbeing are inversely related to poverty. Akelere *et al*, [45] examined the socio-economic factors of household in Ekiti state of Nigeria to control the poverty. Measurements demonstrated that education, household's assets and improved health services negatively are linked with poverty in study area. Ramachandran *et al*., [46] stated that nutritional level and health status could be improved by boosting number of socio-economic factors like education, diet quality and social infrastructure.

Clarke & Erreygers [26] defined and measured health poverty as the situation of being in "poor health" or achieving the worse status than minimally acceptable using the threshold based practices and applied it on three health variables i.e Cardiovascular risk, health status and life expectancy. Analyzed health poverty by employing Self Assessed Health (SAH) technique for Spain and found poor SAH linked with inverse growth of health poverty.

Summing up the above discussion it is observed that numerous socio-economic factors affect the health poverty. The null hypothesis of the study based upon the fact that the selected set of covariates does not affect the multidimensional health poverty, whereas alternative hypothesis is otherwise.

## Methodology

### Specification of data

The study is based upon the secondary cross-sectional data which is collected from the Pakistan Bureau of statistics for the period of 2013–14 to 2018–19. The Federal Bureau of statistics is a sole responsible organization of the Government of Pakistan to conduct the nationwide surveys i.e. HIES, PSLM & PIHS; which are commonly used to estimate the dynamics and correlates of poverty at various levels. The research scholars are issued the survey data upon their requests, while mentioning the motives of the research work and not allowed to present the original data sheets data at any platform.

Methodologically, the current study has been divided into distinct phases; initial part is broadly based on the discussion of indexing measure of multiple attributes of health poverty at

national, provincial and divisional level in Pakistan which is based on seven key dimensions that are briefly described. The second part is based on the detail discussion of decomposition of health poverty with respect to the sub-groups and dimensions. Detail description of key dimensions for the approximation of health poverty is given in **Table 1.**

### Estimation of multidimensional health poverty

To examine the dynamics of health poverty in the multidimensional context, Alkire & Foster [4, 47] initially discussed globally acknowledged and adjusted version of FGT (1984) index. The proposed index is generally grounded on the idea of Amartya Sen's capability approach and gratifies extensive series of axioms projected by Sen [3]. Implementation of such index strongly relies on two experimental steps which are commonly regarded as identification and aggregation phases. The identification step is also disclosed as a dual cut-off method because it is frequently used to categorize poor individuals experiencing the deprivation and poverty threshold. Hence, the first cutoff is known as deprivation cutoff and it is utilized to delineate whether an individual is deprived in the respective dimensions or not. Similarly, second cut-off is described as poverty cutoff and it is employed to decide, whether an individual is to be considered as multidimensional health poor or not. An individual is professed as more than one dimensional poor if his deprivation score $(c_i)$ is more than or equal to a specified poverty yardstick $(k)$ [48]. Alike, the traditional measure of poverty, the calculated outcomes of multi-dimensional poverty are also categorized into further three measures known as adjusted

**Table 1. Profile of dimensional and sub-dimensional framework.**

| Dimensions | Sub-dimensions | Threshold |
|---|---|---|
| $D_1$: Use of Health Services | $d_1$:Consult to doctor during sickness | if person is unable to consult doctor during sickness then 1 otherwise 0 |
| | $d_2$:Assisted Delivery | if women has given birth without taking services of specialized person (doctor, nurse or midwife) then 1 otherwise 0 |
| $D_2$:Quality of Services | $d_3$:Satisfaction with the use of health services | if households is not satisfy with the available health services (govt. and private hospitals) then 1 otherwise 0 |
| | $d_4$:Institutional Delivery | if women has given birth in the house due to unavailability appropriate facility then 1 otherwise 0 |
| $D_3$: Cost of Health Services | $d_5$:Time Cost (Distance) | if more than half hour is required to reach hospital then 1 otherwise 0 |
| | $d_6$: Transport Cost | if an appropriate transport is not available during emergency or too much costly then 1 otherwise 0 |
| $D_4$: Maternal Health | $d_7$:Pre-natal Consultation | if women have no access to prenatal consultation then 1 otherwise 0 |
| | $d_8$:Post Natal Consultation | if women have no access to postnatal consultation then 1 otherwise 0 |
| $D_5$: Child Health | $d_9$:Immunization | if Child has not been immunized then 1 otherwise 0 |
| | $d_{10}$:Checkup | if child did not receive proper checkup from public or private hospital then 1 otherwise 0 |
| | $d_{11}$:Breast feed | if child does not get proper breast feed in initial four months then 1 otherwise 0 |
| $D_6$:Housing Services | $d_{12}$:Safe Sanitation System | if household is victimized of different disease due to poor sanitation system then 1 otherwise 0 |
| | $d_{13}$: Purified Drinker Water | if household has no access to clean drinking water then 1 otherwise 0 |
| $D_7$: Malnutrition | $d_{14}$:Consumption Expenditure on Food | An individual having consumption expenditure on food less then 2350 calories per adult per day in monetary from otherwise 0 |

headcount ratio, adjusted poverty gap ratio and adjusted squared poverty gap ratio or severity of poverty.

## Adjusted headcount ratio

The estimation of adjusted headcount mainly illustrates the proportion of multidimensional health poor. The dual cutoff methodology entails in such a way that the first threshold is described as identification cutoff and the second threshold is known as an aggregation cutoff. Technically, headcount ratio illustrates the magnitude of multidimensional poor across the number of population under consideration. Mathematically, it can be derived as below:

$$H = \frac{q}{n} \tag{1}$$

Empirically, (H) is measured by dividing total deprived individuals (q) in dissimilar dimensions over total population size (n) of study area [49]. Similarly, the second measure of ($A_o$) represents the average deprivation score among deprived individuals in dissimilar dimensions. The general form of particular measure is as follow:

$$A = \sum i(c_i * /d)/q \tag{2}$$

Furthermore, aggregation step comprises of two catalogs commonly named as incidence ($H_o$) and average deprivation gap ($A_o$) of poverty, where product of both these catalogs reflects the adjusted headcount ratio ($M_o$) measure. Hence, the average deprivation gap is calculated as dividing the total sum of deprivations by the total number of population under consideration. However, ($H_o$) is demarcated as a proportion of individuals who are deprived in the multiple ingredients beyond the threshold. Empirically, $M_o$ is defined as follow

$$M_o = H_o \times A_o \tag{3}$$

Finally, the product of headcount ratio and average deprivation gap is collectively known as multidimensional poverty index ($M_o$) [50].

## Adjusted poverty gap ratio

The adjusted poverty gap depicts the proportion of the overall resources to eliminate poverty. Mathematically, it is the product of the measure $M_o$ and G, where 'G' is the average normalized gap to be calculated across the deliberate dimensions. Empirically it is expressed as below:

$$M_1 = M_o \times G$$
$$G = \sum \frac{(\pi_i - v_i)^{\alpha=1}}{\pi_i} \tag{4}$$

## Adjusted squared poverty gap ratio

Likewise, the Adjusted Squared Poverty Gap ratio or adjusted severity of poverty identifies the severity of dispersion among the poor segment of population. In other words, it explains the inequality and estimated by multiplying $M_o$ with S. While 'S' is calculated by taking the square of the each estimated value of G.

$$M_2 = M_o \times S$$
$$S = \sum \frac{(\pi_i - v_i)^{\alpha=2}}{\pi_i} \tag{5}$$

Where, $\pi_i$ is the demonstrative of smallest threshold that differentiate between the deprived and non-deprived in the dissimilar dimensions, while $\upsilon_i$ is the total amount of deprivation across the key dimensions. Finally, like preceding discussion of single dimensional poverty transcript, $\alpha = 0, 1, 2$ are allocated for representation of adjusted intensity and adjusted severity of poverty in the multidimensional spectrum [51].

## Weighting framework

The use of appropriate weighting technique is imperative in the multidimensional spectrum. According to Decancq & Lugo [52], measurement of multiple attribute of poverty largely based upon divergent weighting scheme. However, the weighting methodology mainly depends upon the author choice in the presence of strong literary support. Meanwhile, Khan *et al.*, [53, 54] elaborated that weighting scheme within & across the dimensions has been categorized into two methods commonly regarded as equal and unequal weighting methods. Hence, the lack of strong literary support for unequal weighting method allows the author to use equal weights across and within the dimension [55].

## Results

The estimation of poverty has become a burning debate after the emergence of Millennium Development Goal (MDGs) and Sustainable Development Goals (SDGs). The recent development in the methodology for the approximation of dynamics of poverty has been largely devoted to the seminal work of Alkire & Foster [4]. No doubt, the proposed Multidimensional Poverty Index (MPI) is capable to diagnose the extent of poverty at the sub-national level along the dimensional contribution. Based on the Alkire & Foster [4] methodology, the present study has examined the multidimensional health poverty; thereby employing the set of seven distinct domains from the health spheres. The specified technique of poverty has various advantages. The most common aspect is that it highlights the issue of poverty extensively by using different dimensions at 50% (institutive) dual cut-off as mentioned by Battiston *et al.*, and Alkire & Santos [6, 56, 57] as compared to other multidimensional approaches i.e. Union and Intersection measures as mentioned by Chakravarty *et al.*, [58]. In addition to this, the indexing approach has also other merits like as, it satisfies a wide range of axioms [59] and also decomposes the magnitude of poverty into sub-population as well as dimensional contribution [60]. Similarly, it describes how each sub-group and categorical aspect is contributing to overall multiple health poverty situation of the particular area. Likewise, it also pronounces that how each individual dimension has played its part in determining the multidimensional health deprivation ($H_o$) as well as multidimensional health poverty index ($M_o$). At the national as well as international level, a wide range of literary evidences support the ideology of poverty in the multidimensional context along with dimensional contribution [5, 7, 10, 61, 62].

Table 2 presents the extent of health poverty in the multidimensional spectrum in Pakistan for the last five years on the basis of three nationwide latest cross-sectional data of HIES (Household Integrated Economic Survey) since 2013–14 to 2018–19. The estimated results highlight that at dual cut-off *k = 4*, in the terminal year 2018–19, 33.90% individuals have no access to 4 or more than 4 key dimensions with average deprivation gap of about 61% and intensity, depth and severity of multidimensional health poverty is about 20.80%,16.00% and 14.40% respectively. Like multidimensional poverty statistics as mentioned by Masood *et al.*, & Khan and Akram [63] the present research also demonstrates that health poverty has been purely a rural phenomenon for Pakistan. It has been clearly identified that the rural households were largely victimized of this social issue as compared to urban counterpart. The basic reason behind the higher magnitude of health poverty in the rural segment was lack of access

**Table 2. Profile of multidimensional health poverty in Pakistan: 2013–14 to 2018–19.**

| Groups | Pop. | Cont. | $H_o$ | $A_o$ | $M_o$ | Cont. | $M_1$ | Cont. | $M_2$ | Cont. |
|---|---|---|---|---|---|---|---|---|---|---|
| **Year 2013–14 with Dual Cut-off K = 4** | | | | | | | | | | |
| Punjab | 46681 | 39.22 | 60.90 | 0.701 | 42.70 | 39.50 | 35.80 | 39.78 | 33.30 | 39.70 |
| Urban | 19163 | 48.17 | 53.40 | 0.691 | 36.90 | 50.21 | 30.80 | 50.64 | 28.80 | 50.82 |
| Rural | 27518 | 34.73 | 66.10 | 0.706 | 46.70 | 35.34 | 39.20 | 35.55 | 36.50 | 38.53 |
| Sindh | 33120 | 27.83 | 57.90 | 0.684 | 39.60 | 25.99 | 32.50 | 25.62 | 30.10 | 25.58 |
| Urban | 8007 | 20.13 | 46.60 | 0.678 | 31.60 | 17.97 | 26.00 | 17.86 | 24.30 | 17.92 |
| Rural | 25113 | 31.70 | 61.50 | 0.685 | 42.10 | 29.08 | 34.50 | 28.55 | 32.00 | 28.50 |
| KPK | 25263 | 21.23 | 61.40 | 0.681 | 41.80 | 20.93 | 34.80 | 20.93 | 32.50 | 20.99 |
| Urban | 9099 | 22.88 | 51.50 | 0.672 | 34.60 | 23.85 | 28.40 | 22.18 | 26.50 | 22.21 |
| Rural | 16164 | 20.40 | 67.00 | 0.685 | 45.90 | 20.40 | 38.30 | 20.40 | 35.90 | 17.60 |
| Baluchistan | 13954 | 11.72 | 70.40 | 0.702 | 49.40 | 13.65 | 41.20 | 13.70 | 38.40 | 13.70 |
| Urban | 3515 | 8.83 | 56.00 | 0.682 | 38.20 | 7.34 | 31.20 | 9.40 | 28.70 | 9.28 |
| Rural | 10439 | 13.17 | 75.30 | 0.705 | 53.10 | 15.24 | 44.50 | 15.30 | 41.60 | 15.40 |
| **Pakistan** | **119018** | **100.00** | **61.30** | **0.692** | **42.40** | **100.00** | **35.30** | **100.00** | **32.90** | **100.00** |
| **Urban** | **39784** | **100.00** | **51.80** | **0.683** | **35.40** | **100.00** | **29.30** | **100.00** | **27.30** | **100.00** |
| **Rural** | **79234** | **100.00** | **66.00** | **0.695** | **45.90** | **100.00** | **38.30** | **100.00** | **35.60** | **100.00** |
| **Year 2015–16 with Dual Cut-off K = 4** | | | | | | | | | | |
| Punjab | 62968 | 39.95 | 48.70 | 0.622 | 30.30 | 36.36 | 24.40 | 36.40 | 22.40 | 36.25 |
| Urban | 42718 | 41.46 | 46.70 | 0.623 | 29.10 | 39.30 | 23.50 | 39.45 | 21.60 | 39.30 |
| Rural | 20258 | 37.08 | 52.70 | 0.626 | 33.00 | 32.03 | 26.40 | 31.80 | 24.10 | 31.60 |
| Sindh | 37795 | 23.98 | 52.90 | 0.635 | 33.60 | 24.20 | 27.10 | 24.25 | 25.00 | 24.30 |
| Urban | 22823 | 22.16 | 47.70 | 0.627 | 29.90 | 21.60 | 24.20 | 21.72 | 22.40 | 21.80 |
| Rural | 14972 | 27.41 | 60.90 | 0.647 | 39.40 | 28.30 | 31.60 | 28.12 | 28.90 | 28.00 |
| KPK | 37925 | 24.06 | 56.60 | 0.636 | 36.00 | 26.01 | 29.00 | 26.04 | 26.80 | 26.11 |
| Urban | 23708 | 23.02 | 51.30 | 0.626 | 32.10 | 24.10 | 25.50 | 23.80 | 23.40 | 23.63 |
| Rural | 14217 | 26.02 | 65.50 | 0.650 | 42.60 | 29.02 | 34.80 | 29.40 | 32.40 | 29.80 |
| Baluchistan | 18948 | 12.02 | 55.80 | 0.665 | 37.10 | 13.40 | 30.10 | 13.50 | 27.50 | 13.40 |
| Urban | 13766 | 13.36 | 53.00 | 0.658 | 34.90 | 15.20 | 28.20 | 15.25 | 25.80 | 15.12 |
| Rural | 5182 | 9.49 | 63.20 | 0.682 | 43.10 | 17.71 | 34.90 | 10.75 | 32.00 | 10.73 |
| **Pakistan** | **157636** | **100.00** | **52.50** | **0.634** | **33.30** | **100.00** | **26.80** | **100.00** | **24.70** | **100.00** |
| **Urban** | **103007** | **100.00** | **48.80** | **0.629** | **30.70** | **100.00** | **24.70** | **100.00** | **22.80** | **100.00** |
| **Rural** | **54629** | **100.00** | **59.30** | **0.644** | **38.20** | **100.00** | **30.80** | **100.00** | **28.30** | **100.00** |
| **Year 2018–19 with Dual Cut-off K = 4** | | | | | | | | | | |
| Punjab | 68333 | 42.71 | 29.00 | 0.597 | 17.30 | 35.52 | 13.40 | 35.77 | 12.10 | 35.88 |
| Urban | 22463 | 40.31 | 22.10 | 0.588 | 13.00 | 37.43 | 10.00 | 38.02 | 9.10 | 38.21 |
| Rural | 45870 | 44.00 | 32.50 | 0.596 | 19.40 | 34.98 | 15.00 | 34.92 | 13.60 | 35.20 |
| Sindh | 39050 | 24.41 | 30.70 | 0.609 | 18.70 | 21.95 | 14.20 | 21.66 | 12.80 | 21.70 |
| Urban | 16329 | 29.30 | 18.10 | 0.585 | 10.60 | 22.18 | 8.00 | 22.11 | 7.20 | 22.00 |
| Rural | 22721 | 21.80 | 39.70 | 0.617 | 24.50 | 21.89 | 18.70 | 21.57 | 16.80 | 21.54 |
| KPK | 33814 | 21.14 | 34.80 | 0.615 | 21.40 | 21.75 | 16.50 | 21.80 | 14.90 | 21.88 |
| Urban | 10883 | 19.53 | 24.70 | 0.599 | 14.80 | 20.65 | 11.00 | 20.23 | 9.90 | 20.14 |
| Rural | 22931 | 22.00 | 39.60 | 0.619 | 24.50 | 22.09 | 19.10 | 22.23 | 17.30 | 22.40 |
| Baluchistan | 18784 | 11.74 | 56.90 | 0.641 | 36.50 | 20.60 | 28.30 | 20.77 | 25.50 | 20.80 |
| Urban | 6044 | 10.85 | 40.40 | 0.621 | 25.10 | 19.45 | 19.30 | 19.70 | 17.40 | 19.67 |
| Rural | 12740 | 12.22 | 64.80 | 0.648 | 42.00 | 21.03 | 32.50 | 21.00 | 29.30 | 21.06 |
| Pakistan | 159981 | 100.00 | **33.90** | 0.607 | **20.80** | 100.00 | **16.00** | 100.00 | **14.40** | 100.00 |
| Urban | 55719 | 100.00 | **23.40** | 0.598 | **14.00** | 100.00 | **10.60** | 100.00 | **9.60** | 100.00 |
| Rural | 104262 | 100.00 | **39.60** | 0.616 | **24.40** | 100.00 | **18.90** | 100.00 | **17.00** | 100.00 |

to basic health facilities, distances from BHUs, non-availability of medicine at the time of emergency, high transportation cost and unavailability of experience staff [64, 65]. The results indicate that in urban (rural) regions 23.40% (39.60%) individuals have no access to 4 or more than 4 described attributes with average deprivation gap almost 60% and incidence, depth & severity of multiple dimension based health poverty 14.00%, 10.60% and 9.60% respectively. Furthermore, the provincial level measurements depicted that in the terminal years at overall and regional level Punjab (Baluchistan) was the least (severely) victimized of health poverty as compared to Sindh & KPK. These estimates were consistent with different authors of MDP measurements of health poverty calculations [5, 10, 66]. Like above, Saleem & Khan [67] also elaborated that variation in poverty at provincial level along with regions in Pakistan is due to many common factors.

Finally, the comparative analysis of the study period for five years (2013–14 to 2018–19) highlights that health poverty in Pakistan along with its provinces & regions in the multidimensional view point has declined significantly. However, at the national level along with regions (urban & rural) the precise decline in health poverty was estimated around 51% (60.45% & 46.85%) while at provincial level in Punjab, Sindh, KPK and Baluchistan, the particular down trend in poverty was almost 59.48% (64.76% & 58.45%); 52.80% (94.93% & 41.80%); 48.80% (57.22% & 46.62%) & 26.11% (34.30% & 20.90%) respectively. The findings comprehensively demonstrated that over the period of time especially with 5 years gap, a considerable downfall in the health poverty at national, provincial and regional level has been experienced in Pakistan.

The decomposition of multidimensional health poverty across all the four provinces of Pakistan since 2013–14 to 2018–19 has been presented in Table 3. The decomposition analysis elucidates that how much the population of each province and its regions in the absolute (relative) term has contributed to the overall statistics of multidimensional health poverty. The calculations of decomposition quite persuasively exposed that all the provinces along with

**Table 3. Decomposition analysis of multidimensional health poverty: 2013–14 to 2018–19.**

| Regions | 2013–14 | | | 2015–16 | | | 2018–19 | | |
|---|---|---|---|---|---|---|---|---|---|
| | $M_o$ | A.C.[*] | R.C.[**] | $M_o$ | A.C.[*] | R.C.[**] | $M_o$ | A.C.[*] | R.C.[**] |
| Punjab | 42.70 | 16.80 | 39.83 | 30.30 | 12.70 | 37.39 | 17.30 | 7.30 | 38.05 |
| Urban | 36.90 | 18.20 | 50.11 | 29.10 | 13.40 | 40.30 | 13.00 | 5.40 | 39.63 |
| Rural | 46.70 | 15.50 | 35.45 | 33.00 | 12.40 | 32.35 | 19.40 | 9.00 | 37.40 |
| Sindh | 39.60 | 10.50 | 26.07 | 33.60 | 7.50 | 24.30 | 18.70 | 5.30 | 23.01 |
| Urban | 31.60 | 6.90 | 19.42 | 29.90 | 6.70 | 22.32 | 10.60 | 3.50 | 27.72 |
| Rural | 42.10 | 12.60 | 28.91 | 39.40 | 10.20 | 26.99 | 24.50 | 5.20 | 21.05 |
| KPK | 41.80 | 9.40 | 21.22 | 36.00 | 8.60 | 27.08 | 21.40 | 4.50 | 21.06 |
| Urban | 34.60 | 6.80 | 21.77 | 32.10 | 7.10 | 24.79 | 14.80 | 3.10 | 17.84 |
| Rural | 45.90 | 10.90 | 20.99 | 42.60 | 11.60 | 31.04 | 24.50 | 5.60 | 22.40 |
| Baluchistan | 49.40 | 5.70 | 12.88 | 37.10 | 4.50 | 11.50 | 36.50 | 3.70 | 17.87 |
| Urban | 38.20 | 3.60 | 8.70 | 34.90 | 3.50 | 12.59 | 25.10 | 2.00 | 14.80 |
| Rural | 53.10 | 6.90 | 14.66 | 43.10 | 4.00 | 9.62 | 42.00 | 4.60 | 19.15 |
| **Pakistan** | **42.40** | **42.40** | **100.00** | **33.30** | **33.30** | **100.00** | **20.80** | **20.80** | **100.00** |
| **Urban** | **35.40** | **35.40** | **100.00** | **30.70** | **30.70** | **100.00** | **14.00** | **14.00** | **100.00** |
| **Rural** | **45.90** | **45.90** | **100.00** | **38.20** | **38.20** | **100.00** | **24.40** | **24.40** | **100.00** |

[*]Absolute Contribution

[**] Relative Contribution

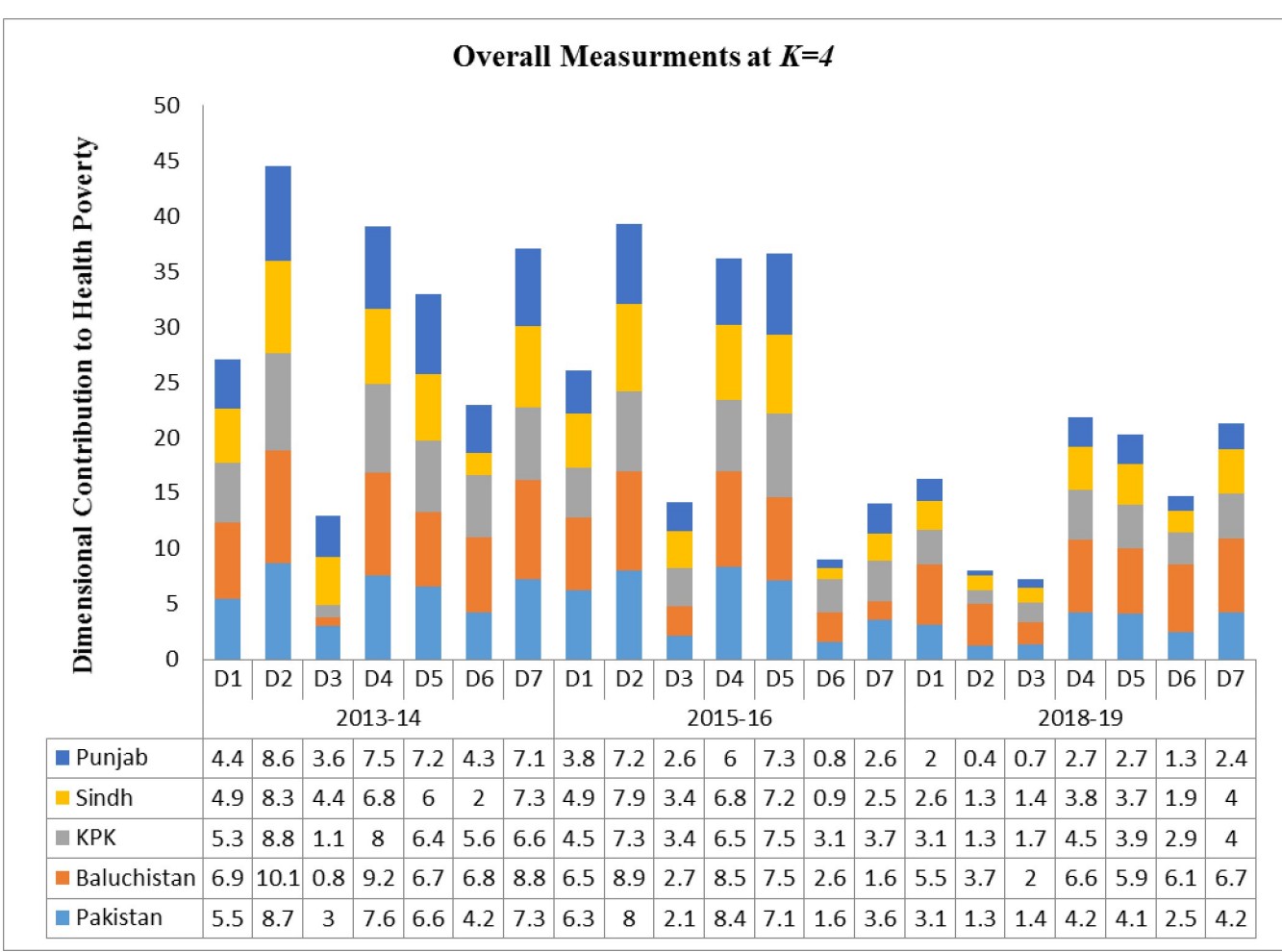

**Fig 1. Dimensional decomposition of multidimensional health poverty (M$_o$) index through Shapley approach.**

regions have significantly contributed to the overall Pakistan and its regions health poverty index. However, two provinces Punjab & Sindh, due to higher population as mentioned in **Table 2** were the major contributor to overall national poverty index in absolute (relative) term across all the study years.

The estimations of multidimensional health poverty (M$_o$) have also been validated by the dimensional contribution under the Shapley method of decomposition. The Shapley approach has the advantage to measure the population as well as dimensional contribution; whereas the other methodologies only encircle the population contribution in the absolute & relative terms. In addition to this, the Shapley approach based upon the simple computation [68]. Calculations of overall country and all the provinces demonstrate that all seven dimensions; use of health service, quality of health service, cost of health service, maternal health, child health, housing services associated with health and malnutrition have considerable contribution to the multidimensional health poverty index in Pakistan (Fig 1). Moreover, comparative analysis of dimensions across all three study period 2013–14, 2015–16 & 2018–19 demarcate that five key dimensions namely; uses of health services, quality of health services, maternal health, child health and malnutrition were the major contributor to multidimensional health poverty in study area. Nevertheless, over the period of time since 2013–14 to 2018–19 it has also been

perceived that contribution of most of indicator has been significantly decline at country and provincial level in Pakistan. These assessments also certify above views that over the time development in health sector has been taken place that leads to lessen the deprivation of residents of all the provinces at overall as well as regional level that leads to decline multiple attributes health poverty at country level in Pakistan along with urban & rural regions.

## Discussion

The dynamics of health poverty has been estimated in the multidimensional spectrum at the national, provincial, divisional and regional levels in Pakistan by employing the Alkire *et al.*, [4] methodology. Because of the fact, that given measure of poverty has many advantages and has considerable supremacy due to various aspects over the different methods. For instance, the most common feature of the proposed methodology is that at one side it highlights the issue of poverty extensively by using different dimensions.

Interestingly, the results indicate the precise decline in health poverty in the urban regions was greater than remote areas which means over time improvement in basic health services in Pakistan has took place with greater extent in urban areas as compared to rural masses. Like income poverty, hunger & education, the adequate health status has also been the key indicator of SDGs (Sustainable Development Goal) derived by the UNDP (United Nation Development Program); thus, the reduction in health poverty till end of 2030 has keen crucially important. Estimated measurements authenticate that if economy of Pakistan continues current consistency in health sector by providing easy basic health services to the needy people especially residing in rural areas, then it would be a great chance to achieve SDG of zero health poverty by the end of upcoming decade. Finally, the comparative analysis of provinces highlighted that reduction in poverty in the province of Baluchistan was considerably lower as compared to all others sub-nationals, therefore it is necessary for the government to take emergency steps for the betterment of residence of Baluchistan province in the health sector by providing health insurance as well as key health services like basic vaccines for children & availability of experience staff along with essentially required medicine for females & children at their door step.

Additionally, alike multidimensional health poverty, it has also been quite authentically noted that over the period of time precise contribution has been substantially reduced across all the provinces & their regions, which means that boosting health services and introducing latest technology in health sector at provincial level led to decline the health poverty that ultimately has mitigated the overall statistics. On the other hand, it is the need of the time to meet the SDG,s i.e. zero poverty in the milieu of health till 2030. Therefore, it is imperative to maintain the adequate consistency to decline the poverty statistics till the end of next decade.

## Conclusion

In the history of developing economies, the poverty debate had largely based on the conventional approaches. It is well known fact that the traditional ideology defines poverty in the single dimensional context by using income or consumption expenditure as basic indicator and $ 1.25/ day per individual as minimum yardstick [12, 69, 70]. The recent conceptual development designated poverty as a multidimensional dilemma by using three key globally accepted social attributes education, health and housing services along with dual cut off *k = 2* as minimum benchmark [7, 48, 71]. With the passage of time, the multidimensional poverty debate has emerged with the idea of multidimensional health poverty [10]. Due to the importance of particular social issue, current study has made a pioneering effort to examine the health poverty in the multidimensional context at national, provincial and regional level in Pakistan by using the seven key dimensions through the universally accepted Alkire and Foster [4]

indexing method and latest available data of HIES (Household Integrated Economic Survey) for last five years since 2013–14 to 2018–19. All the numerical outcomes clearly identify that over the period of time; especially a significant decline in health poverty at national, provincial and regional level in Pakistan has been observed during 5 year gap. Though this decreasing trend in health poverty in the rural region was found relatively greater than urban areas which clearly indicates a larger extent of improvements in basic health services over the time, yet estimated outcomes authenticate that similar consistency of political stability along with the same thrust of remedial measures in the health sector till the end of 2030, may be more beneficial for the emerging economy of Pakistan to achieve SDG of zero health poverty within the stipulated period. Finally, the comparative analysis of provinces signify that reduction in poverty in the Baluchistan province was significantly lower as compared to all others provinces, therefore it is necessary for the government to take war footed steps for the betterment of the resident of Baluchistan in health sector by providing health insurance and also key health services like basic vaccines for children & availability of experienced staff along with essentially required medicine particularly for the more deprived segments of the province.

## Implications

In the light of estimated outcomes, following policy lesson can be derived to overcome the problem of health poverty more effectively and efficiently:

- There is a dire need to spread the net of health cards across all the provinces; thereby boosting the health consciousness and easy accessibility of health services, particularly for the poor & deprived masses of population.

- In the remote areas, immunization of children is a serious challenge for people especially in the provinces of KPK & Baluchistan, therefore it is the prime responsibility of government to take emergency steps for the availability of accessible immunization services at the door step of needy people of the particular provinces.

- It is necessary for the government to take emergency steps to ensure the provision of basic health services for females of rural areas especially pre-natal & post-natal and assisted birth facilities along with required trained staff.

## Limitations

There are many advantages of current data set taken under consideration like it is easy to handle, low margin of errors, easily breakable at national, provincial, divisional, regional and district level, easily available and also hold detail information of overall country regarding most of the social issues. Among these advantages, the particular data set also has some of the drawbacks, which can never be ignored while calculating the results of various social issues like poverty, inequality pro poor growth etc. Initially, its collection is very dull which means in the year 2021 the data of HIES is available at year 2018–19 and since 2014–15 no data set of PSLM has been collected yet. Secondly, its collection is also too much costly as well as difficult due to non-availability of various basic resources like availability of easy transport in difficult hilly areas, experienced field staff etc. However, sample size of data set is also quite small as compared to country overall population. Therefore, a precise sample selection method is needed to be revised on quick bases for the attainment of more logical outcomes. Finally, data collection pattern is not appropriate few data sets are collected with 1 year gap some are collected with 2 or 3 years gaps such inconsistency makes estimation of the overall study doubtful that generates serious complications in policy making and implementation.

On the whole, despite having these limitations, it is one the very rich & informative data source and frequently used for most of the socio-economic analysis. An additional positive aspect of this data set is that it contains a largest sample size; which by default resolve most of the issues technically.

## Recommendations

- It is a need of the time to improve health services by introducing modern technology especially in the most affected remote areas; so that the poor and needy people may get easy access to the modern health services.

- There is dare need to aware the people living in the rural areas about the benefits of precise measures through mass and print media.

- In the health sector, unfortunately the unavailability of clean drinking water and the absence of consultant along with appropriate medicine are the key challenges that cause deadly diseases especially in women and children like typhoid, diarrhea, tetanus and hepatitis etc. Therefore to minimize these problem governments may take emergency steps in the provision of both these facilities toward the individuals of deprived areas especially residing in town, mouzas and dehs in Pakistan.

## Author Contributions

**Conceptualization:** Kiran Mustafa, Misbah Nosheen, Atta Ullah Khan.

**Data curation:** Kiran Mustafa.

**Methodology:** Kiran Mustafa, Misbah Nosheen, Atta Ullah Khan.

**Supervision:** Misbah Nosheen, Atta Ullah Khan.

**Writing – original draft:** Kiran Mustafa.

**Writing – review & editing:** Misbah Nosheen, Atta Ullah Khan.

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
