## [Decision Letter · Decision Letter 0]

27 Jun 2021

PONE-D-21-09200

Dynamic Reflections of Multidimensional Health Poverty in Pakistan

PLOS ONE

Dear Dr. Khan,

Thank you for submitting your manuscript to PLOS ONE. After careful consideration, we feel that it has merit but does not fully meet PLOS ONE’s publication criteria as it currently stands. Therefore, we invite you to submit a revised version of the manuscript that addresses the points raised during the review process.

The manuscript investigates a very interesting topic of multidimensional health poverty. To improve its quality, reviewers have made suggestions particularly to strengthen the sections of introduction, methodology and discussions (see review reports).

We look forward to receiving your revised manuscript.

Kind regards,

Abid Hussain

Academic Editor

PLOS ONE

Journal Requirements:

3. Please include your tables as part of your main manuscript and remove the individual files. Please note that supplementary tables (should remain/ be uploaded) as separate "supporting information" files.

4. Please ensure that you include a title page within your main document. We do appreciate that you have a title page document uploaded as a separate file, however, as per our author guidelines (http://journals.plos.org/plosone/s/submission-guidelines#loc-title-page) we do require this to be part of the manuscript file itself and not uploaded separately.

6. We note you have included a table to which you do not refer in the text of your manuscript. Please ensure that you refer to Table 3 in your text; if accepted, production will need this reference to link the reader to the Table.

Reviewers' comments:

Reviewer's Responses to Questions

**Comments to the Author**

1. Is the manuscript technically sound, and do the data support the conclusions?

Reviewer #1: Yes

Reviewer #2: Yes

2. Has the statistical analysis been performed appropriately and rigorously? 

Reviewer #1: Yes

Reviewer #2: Yes

3. Have the authors made all data underlying the findings in their manuscript fully available?

Reviewer #1: Yes

Reviewer #2: Yes

4. Is the manuscript presented in an intelligible fashion and written in standard English?

Reviewer #1: Yes

Reviewer #2: Yes

5. Review Comments to the Author

Reviewer #1: Title: Dynamic Reflections of Multidimensional Health Poverty in Pakistan

Manuscript ID: PONE-D-21-09200

Referee Recommendation: MINOR REVISIONS

I appreciate the author(s) for conducting this study on multidimensional health poverty in Pakistan via a new and novel way, especially the inter-provincial comparison. However, I have noted some issues which may render further quality to the paper, as following:

Comments:

1. I would suggest the author to cite most recent literature.

2. I would like to ask why equal weighting technique has been adopted instead of unequal?

3. How the cutoff decision has been made. Please mention the reference.

4. I would suggest the author to mention the hypothesis in revised manuscript.

5. Briefly define health poverty in introduction section to enhance the clarity for readers.

6. Limitations of this study should be incorporated in revised manuscript.

7. How this study is different from Iqbal & Nawaz (2017) study as it seems the replication of that study!

8. Individuals’ health poverty is directly judged by health status of individuals then why such an important indicator has not been addressed in this study?

9. Different dimensions like expenditure on food and sanitation cannot have equal importance in determining individuals’ health then why these two are equally weighted?

10. What is the justification of using Shapley approach? And not been defined theoretically.

11. It is suggested to explain dimensional decomposition.

12. Policy recommendation section needs improvement.

13. Page # 3, Line# 2: Change the word effect to affect.

Reviewer #2: First, I would like to thank you to offer this exciting opportunity to assess this study. I believe the authors have examined an interesting research topic, entitled, "Dynamic Reflections of Multidimensional Health Poverty in Pakistan." I have enjoyed reading and evaluating this article as it matches my research interests. This article describes its objective to explore, analyze and evaluate multidimensional health poverty in the country.

The study describes that recent methodological development has entirely shifted the identification of poor in the multidimensional spectrum; thereby addressing the multiple health spheres. The present research primarily examines the dynamics of multidimensional health poverty because of HIES & PSLM nationwide survey data from 2013-14 to 2018-19. The study employed Alkire & Foster (2007) Multidimensional Poverty Index to estimate the seven distinct dimensions of health aspects to identify the poor.

The study has focused on a critical issue of health aspects that is top priority issue of the society. I have some suggestions for the authors to enhance this work quality. I am recommending your research for publication. It is a good topic; however, you need to work on my suggestions to reach scientific merit. Make changes one by one as suggested. First, your flow of the study is not appropriate, especially, methods, results etc. Please each section and follow the recommendations to improve your study. Sequence should like this; (1) Introduction, (2) Literature, (3), Methods, (4) Results, (5) Discussion, (6) Conclusion, (7) Implications, (8) Limitations, (9) Recommendations. Follow guidelines and work on systematically. I have accepted your study for publication, but you need to work on my suggestions one by one.

Abstract

Revise the abstract and cover it with a theme reflecting the central idea of the study. The abstract is the face of main study. It must be high quality. Make improvement and fix typo errors.

Introduction section

The introduction needs improvement. Discuss research gaps identified from the literature. In my opinion, points could very interesting in this study. How can social media play a leading role in educating people about this health issue? How CSR and knowledge sharing can improve the situation during the COVID-19 challenging situation?

Hussain, T., Abbas, J., Wei, Z., Ahmad, S., Xuehao, B., & Gaoli, Z. (2021). Impact of Urban Village Disamenity on Neighboring Residential Properties: Empirical Evidence from Nanjing through Hedonic Pricing Model Appraisal. Journal of Urban Planning and Development, 147(1), 04020055. doi:10.1061/(asce)up.1943-5444.0000645

Shuja, K. H., Shahidullah, Aqeel, M., Khan, E. A., & Abbas, J. (2020). Letter to highlight the effects of isolation on elderly during COVID-19 outbreak. Int J Geriatr Psychiatry, n/a(n/a). doi:10.1002/gps.5423

Abbas, J., Zhang, Q., Hussain, I., Akram, S., Afaq, A., & Shad, M. A. (2020). Sustainable Innovation in Small Medium Enterprises: The Impact of Knowledge Management on Organizational Innovation through a Mediation Analysis by Using SEM Approach. Sustainability, 12(6), 2407. doi:https://doi.org/10.3390/su12062407

I strongly suggest you build your study with the idea. Your research will become excellent as you have already explored a great idea. The whole world is facing such issues in society; however, it is becoming common in some developing countries. These rich people can spend this wealth for the social good to become immortal. I am suggesting outstanding studies published in leading journals. Please read these studies, improve your introduction, and follow these articles to enhance your work quality.

Azizi, M. R., Atlasi, R., Ziapour, A., Abbas, J., & Naemi, R. (2021). Innovative human resource management strategies during the COVID-19 pandemic: A systematic narrative review approach. Heliyon, 6(12).

Su, Z., McDonnell, D., Wen, J., Kozak, M., Abbas, J., Šegalo, S., Li, X., Ahmad, J., Cheshmehzangi, A., Cai, Y., Yang, L., & Xiang, Y.-T. (2021, 2021/01/05). Mental health consequences of COVID-19 media coverage: the need for effective crisis communication practices. Globalization and Health, 17(1), 4. https://doi.org/10.1186/s12992-020-00654-4

Abbas, J. (2020). The Impact of Coronavirus (SARS-CoV2) Epidemic on Individuals Mental Health: The Protective Measures of Pakistan in Managing and Sustaining Transmissible Disease. Psychiatr Danub, 32(3-4), 472-477. doi:10.24869/psyd.2020.472

Aqeel, M., Abbas, J., Shuja, K. H., Rehna , T., Ziapour, A., Yousaf, I., & karamat, T. (2020). The Influence of Illness Perception, Anxiety and Depression Disorders on Students Mental Health during COVID-19 Outbreak in Pakistan: A Web-Based Cross-Sectional Survey. International Journal of Human Rights in Healthcare, 14.

Yoosefi Lebni, J., Abbas, J., Moradi, F., Salahshoor, M. R., Chaboksavar, F., Irandoost, S. F., Nezhaddadgar, N., & Ziapour, A. (2020, Jul 2). How the COVID-19 pandemic effected economic, social, political, and cultural factors: A lesson from Iran. International Journal of Social Psychiatry, 20764020939984. https://doi.org/10.1177/0020764020939984

Literature section:

The literature section needs improvement. I suggest the authors to look into the suggested studies to improve literature section. Build your idea how innovative strategies can bring change in the government organizations to address this problem. The authors add the latest citations to the literature and method sections to enhance the study's quality. Follow these studies in the literature to enhance the quality of your work.

NeJhaddadgar, N., Ziapour, A., Zakkipour, G., Abbas, J., Abolfathi, M., & Shabani, M. (2020). Effectiveness of telephone-based screening and triage during COVID-19 outbreak in the promoted primary healthcare system: a case study in Ardabil province, Iran. Z Gesundh Wiss, 1-6. doi:10.1007/s10389-020-01407-8

Abbas, J. (2021). Crisis management, transnational healthcare challenges and opportunities: The intersection of COVID-19 pandemic and global mental health. Research in Globalization, 100037. https://doi.org/10.1016/j.resglo.2021.100037

Abbas, J., Aman, J., Nurunnabi, M., & Bano, S. (2019). The Impact of Social Media on Learning Behavior for Sustainable Education: Evidence of Students from Selected Universities in Pakistan. Sustainability, 11(6), 1683. doi:10.3390/su11061683

Maqsood, A., Abbas, J., Rehman, G., & Mubeen, R. (2021). The paradigm shift for educational system continuance in the advent of COVID-19 pandemic: Mental health challenges and reflections. Current Research in Behavioral Sciences, 2, 100011. doi:https://doi.org/10.1016/j.crbeha.2020.100011

Abbas, J., Wang, D., Su, Z., & Ziapour, A. (2021). The Role of Social Media in the Advent of COVID-19 Pandemic: Crisis Management, Mental Health Challenges and Implications. Risk Manag Healthc Policy, Volume 14, 1917-1932. https://doi.org/10.2147/rmhp.S284313

Su, Z., McDonnell, D., Cheshmehzangi, A., Abbas, J., Li, X., & Cai, Y. (2021). The promise and perils of Unit 731 data to advance COVID-19 research. BMJ Global Health, 6(5), e004772. https://doi.org/10.1136/bmjgh-2020-004772

Methods section

Make a proper heading of the method section and explain it. I suggest adding demographic table by covering education level, age, income level and regions. See the suggested study and improve your study. See these studies to improve your work and follow them in the methods and results sections.

Mubeen, R., Han, D., Abbas, J., & Hussain, I. (2020). The Effects of Market Competition, Capital Structure, and CEO Duality on Firm Performance: A Mediation Analysis by Incorporating the GMM Model Technique. Sustainability, 12(8), 3480. doi:10.3390/su12083480

Su, Z., Wen, J., Abbas, J., McDonnell, D., Cheshmehzangi, A., Li, X., . . . Cai, Y. (2020). A race for a better understanding of COVID-19 vaccine non-adopters. Brain Behav Immun Health, 9, 100159. doi:10.1016/j.bbih.2020.100159

Local Burden of Disease, H. I. V. C. (2021, Jan 8). Mapping subnational HIV mortality in six Latin American countries with incomplete vital registration systems. BMC Medicine, 19(1), 4. https://doi.org/10.1186/s12916-020-01876-4

Abbas, J., Aqeel, M., Ling, J., Ziapour, A., Raza, M. A., & Rehna, T. (2020). Exploring the relationship between intimate partner abuses, resilience, psychological, and physical health problems in Pakistani married couples: a perspective from the collectivistic culture. Sexual and Relationship Therapy, 1-30. doi:10.1080/14681994.2020.1851673

Results section

See the suggested study and improve your results section. You can add graphical presentation of your findings. See these studies to improve your study.

Hussain, T., Abbas, J., Wei, Z., & Nurunnabi, M. (2019). The Effect of Sustainable Urban Planning and Slum Disamenity on The Value of Neighboring Residential Property: Application of The Hedonic Pricing Model in Rent Price Appraisal. Sustainability, 11(4), 1144. doi:10.3390/su11041144

Abbas, J., Hussain, I., Hussain, S., Akram, S., Shaheen, I., & Niu, B. (2019). The Impact of Knowledge Sharing and Innovation upon Sustainable Performance in Islamic Banks: A Mediation Analysis through an SEM Approach. Sustainability, 11(15), 4049. doi:10.3390/su11154049

Abbas, J., Raza, S., Nurunnabi, M., Minai, M. S., & Bano, S. (2019). The Impact of Entrepreneurial Business Networks on Firms’ Performance Through a Mediating Role of Dynamic Capabilities. Sustainability, 11(11), 3006. doi:10.3390/su11113006

Discussion section:

Make some improvement in the discussion section. It should be around one page and a half. Make it strong. See the recommended studies, and improve your sections.

Conclusion

Make a separate heading for conclusion and do not mix it with the discussion section. Your conclusion should be based on minimum 500 words.

Implications

Make a separate heading for implications of your study. I suggest adding implications heading and briefly explain it in this section.

Limitations

Make a proper heading and discuss it adequately.

Highlight the study’s scientific contribution to scientific knowledge. The authors should explain how this study offers useful insights to the researchers of the tourism industry in the discussion section. The English level needs corrections to meet scientific merit for publication. I accept and endorse this manuscript for publication after minor modifications, as suggested.

6. PLOS authors have the option to publish the peer review history of their article (what does this mean?). If published, this will include your full peer review and any attached files.

Reviewer #1: No

Reviewer #2: No

---

## [Author Response · Author response to Decision Letter 0]

28 Sep 2021

Reviewer-1

1. Incorporated as suggested accordingly. (pg-6)

2. In the absence of strong literary evidence the researcher is allowed to opt the equal weighting strategy and similar formation has been adopted by earlier author like Iqbal & Nawaz , (2017). (pg-9)

3. It has been described in different spots in literature review that’s why not included in introduction to avoid the replication. (pg-3)

4. Hypothesis is the optional part of the research, however, the null hypothesis of the study based upon the fact that the selected set of covariates correlates does not affect the multi dimensional health poverty, whereas alternative hypothesis is otherwise. (pg-6)

5. The intuitive cut off strategy has been adopted by as proposed by Alkire and Foster (2007), Alkire and Santos (2010) and Alkire and Housseini (2014) (pg-7)

6. Equal weighting strategies has been adopted as suggested. (pg-9)

7. The study in hand is entirely based upon HIES and PSLM data, despite the fact many other health aspects may isturb the magnitude of multidimensional health poverty. Though the study in hand is based upon HIES AND PSLM data but it is extended to the recent data along with the unique set of data. (pg-14)

8. Because of the limited information in the existing data the household health poverty has been estimated as a representative of individual health status. (pg- 5)

9. These aspects has been taken as an indicator to present the nutritional deprivations of individuals. (pg-4)

10. Justified the Shapley approach accordingly and added literary justification. (pg-8)

11. Incorporated as suggested. (pg-11)

12. Incorporated as suggested. (pg-11)

13. Incorporated as suggested. (pg-14)

Reviewer-2

Abstract: Revised accordingly and improved the typo mistakes. (pg-1)

Introduction section: Incorporated as suggested. (pg-3)

Methods section: Improved accordingly. (pg- 6-9)

Results section: Reviewed and incorporated the suggested studies. (pg-10-13)

Discussion section:: Improved accordingly. (pg-12-13)

Conclusion: Corrected as per suggestions. (pg-13)

Implications: Corrected as per suggestions. (pg-15)

Limitations: Corrected in the light of suggestions. (pg-14-15)

---

## [Editor Report · Decision Letter 1]

11 Oct 2021

Dynamic Reflections of Multidimensional Health Poverty in Pakistan

PONE-D-21-09200R1

Dear Dr. Khan,

We’re pleased to inform you that your manuscript has been judged scientifically suitable for publication and will be formally accepted for publication once it meets all outstanding technical requirements.

Kind regards,

Abid Hussain

Academic Editor

PLOS ONE
---

## [Editor Report · Acceptance letter]

25 Oct 2021

PONE-D-21-09200R1 

Dynamic Reflections of Multidimensional Health Poverty in Pakistan 

Dear Dr. Khan:

I'm pleased to inform you that your manuscript has been deemed suitable for publication in PLOS ONE. Congratulations! Your manuscript is now with our production department. 

Kind regards, 

on behalf of

Dr. Abid Hussain 

Academic Editor

PLOS ONE